# Parkinson’s Disease: Cells Succumbing to Lifelong Dopamine-Related Oxidative Stress and Other Bioenergetic Challenges

**DOI:** 10.3390/ijms25042009

**Published:** 2024-02-07

**Authors:** Hirohisa Watanabe, Johannes M. Dijkstra, Toshiharu Nagatsu

**Affiliations:** 1Department of Neurology, School of Medicine, Fujita Health University, Toyoake 470-1192, Aichi, Japan; 2Center for Medical Science, Fujita Health University, Toyoake 470-1192, Aichi, Japan; 3Center for Research Promotion and Support, Fujita Health University, Toyoake 470-1192, Aichi, Japan; tnagatsu@fujita-hu.ac.jp

**Keywords:** Parkinson’s disease, substantia nigra, dopamine, neuromelanin, α-synuclein, oxidative stress, mitochondria, calcium, energy, ATP

## Abstract

The core pathological event in Parkinson’s disease (PD) is the specific dying of dopamine (DA) neurons of the substantia nigra pars compacta (SNc). The reasons why SNc DA neurons are especially vulnerable and why idiopathic PD has only been found in humans are still puzzling. The two main underlying factors of SNc DA neuron vulnerability appear related to high DA production, namely (i) the toxic effects of cytoplasmic DA metabolism and (ii) continuous cytosolic Ca^2+^ oscillations in the absence of the Ca^2+^-buffer protein calbindin. Both factors cause oxidative stress by producing highly reactive quinones and increasing intra-mitochondrial Ca^2+^ concentrations, respectively. High DA expression in human SNc DA neuron cell bodies is suggested by the abundant presence of the DA-derived pigment neuromelanin, which is not found in such abundance in other species and has been associated with toxicity at higher levels. The oxidative stress created by their DA production system, despite the fact that the SN does not use unusually high amounts of energy, explains why SNc DA neurons are sensitive to various genetic and environmental factors that create mitochondrial damage and thereby promote PD. Aging increases multiple risk factors for PD, and, to a large extent, PD is accelerated aging. To prevent PD neurodegeneration, possible approaches that are discussed here are (1) reducing cytoplasmic DA accumulation, (2) blocking cytoplasmic Ca^2+^ oscillations, and (3) providing bioenergetic support.

## 1. Introduction

Parkinson’s disease (PD) is defined as an age-related, clinically evident Parkinsonism, and pathologically neurodegenerative disease with specific neuronal loss of dopamine (DA) neurons in the substantia nigra pars compacta (SNc) and norepinephrine (NE) neurons in the locus coeruleus (LC) that are rich in neuromelanin (NM), a decrease in NM, and formation of toxic misfolded oligomers of α-synuclein [1]. At least temporarily, treatment with L-DOPA (also known as “levodopa”), which provides a replacement for decreasing natural DA, can ameliorate symptoms but not the progression of the disease [2]. Monoamine oxidase type B (MAO-B) inhibitors, catechol-O-methyl-transferase (COMT) inhibitors, and deep brain stimulation therapy are also widely used, but these treatments cannot stop the progression of the disease either [3]. Even in cell transplantation therapy, Lewy bodies have also been reported to appear in grafts, and the value of the therapy remains to be seen [4,5,6].

The most significant risk factor for PD is aging. It is estimated that the number of PD patients worldwide doubled to more than 6 million between 1990 and 2015, mainly due to aging, and it is projected to exceed 12 million by 2040 [7]. PD, which is essentially the preferential dying of DA neurons, is believed to be caused by the sensitivity of those neurons to harmful endogenous and/or exogenous factors. Those factors can vary, be singular or multiple, and are the topic of much debate [8,9]. The present review focuses on two main underlying reasons causing SNc DA neurons’ specific vulnerability: (i) their bioenergetic demands and (ii) their high DA production. It also provides a historical perspective, showing that whereas initially, the bioenergetic demands were simply portrayed as “high metabolism”, gradually, a more nuanced understanding of mitochondria under particular oxidative stress emerged. We discuss how several risk factors appear to act on PD vulnerabilities and how understanding these vulnerabilities may lead to strategies against the disease.

## 2. Progression of PD Pathology

### 2.1. Normal Age-Related Loss of SNc DA Neurons

Even in healthy aging, SNc DA neurons show their specific vulnerability by substantially higher losses (roughly 5–10% per decade) than many other types of neurons [10]. This is probably caused by several risk factors that increase with aging, as discussed below. PD, with few exceptions, is a disease of the elderly and, to a large extent, appears to involve an acceleration/worsening of the normal age-related deterioration of the nigrostriatal DA system. The healthy nigrostriatal DA system seems to provide considerable surplus DA for maintaining “normal” motor functioning. It is commonly estimated that only when nigral DA neuron counts and striatal DA levels are diminished by about 50% and 80%, respectively, Parkinsonian motor symptoms appear [11,12,13]. However, it should be considered that some motor function abnormalities tend to already appear several years before classic PD symptoms manifest [14]. Furthermore, in a discussion on how much DA is necessary for normal function, it should be realized that the brain appears to have compensatory mechanisms for dealing with some levels of DA deficiencies [15].

### 2.2. The Braak Model of PD Staging

SNc DA neuron vulnerability and enhanced degeneration are common aspects among PD cases, but the triggers and anatomical spreading routes of PD vary [9]. In most—although not all—cases, the progression of PD includes the generation of Lewy bodies (LB), which are aggregates that have α-synuclein proteins as their primary components [16], and toxic α-synuclein oligomers are believed to spread as prions between neurons [17]. LBs are also increased in healthy aging and do not always lead to PD or other distinguishable neurological diseases [18]. However, LBs, on average, are significantly enhanced in PD, and a frequently observed sequential order of their appearance in different neuronal regions in postmortem PD brains led Braak and co-workers to propose what has become known as the “Braak hypothesis” or “dual-hit hypothesis” for PD staging [19,20]. The dual-hit hypothesis postulated that the pathology—potentially induced by a neurotropic pathogen—enters the brain simultaneously via a nasal and a gastric route, based on the observations of initial LB lesions in “stage 1” PD in the olfactory bulb, anterior olfactory nucleus, the dorsal nucleus of the vagus nerve (which connects with the gastrointestinal tract), and the intestine and implies retrograde transport from the environment to the central brain [21]. However, later studies have proposed the single-hit hypothesis (brain-first PD or body-first PD) [22,23] since few PD cases simultaneously present with peripheral nerve and olfactory tract lesions. Most cases can be divided into two types: one with lesions localized to the peripheral autonomic nervous system and the other with lesions localized to the olfactory tract [22]. Based on the location of the initial LB lesions, the outside world seems to be a logical suspect for being the instigator of PD in both the dual-hit and single-hit hypotheses.

### 2.3. A Contributing Role of Lewy Bodies

Roughly, only about half of PD cases present the pattern of occurrence of LB consistent with the Braak hypothesis [24], and it has been shown that many cellular and molecular neuropathological changes occur before the appearance of LB [25]. It should also be noted that some forms of familial PD, such as those resulting from mutations in LRRK2 [26] or PARK2 [27], lack a common association with LB. Furthermore, in a study of idiopathic PD brains, the severity of dorsal vagal nucleus lesions did not correlate with the severity of cortical lesions in semiquantitative assessments [28]. Overall, no theory has yet been proposed that can explain all of the pathophysiology of LB.

Because LB can be found without PD and vice versa, it is tempting to speculate that LB is only a bystander phenomenon and is not the direct cause of PD. However, multiple different mutations in the gene for α-synuclein, *SNCA*, are associated with familial PD [29], providing evidence that α-synuclein-related phenomena can directly contribute to PD. The most straightforward way to interpret all observations is that LB/α-synuclein spreading is one of the factors that can contribute to PD but does not always lead to PD and is not necessary for inducing PD. A crucial aspect of the Braak model is the realization that PD can start in different brain regions and may pathologically spread toward the SNc from there.

### 2.4. Cellular and Regional Differences within the SNc

The SNc is not homogeneous, and not all SNc neurons die equally in PD. They are mostly the dopaminergic neurons—identified by being positive for tyrosine hydroxylase (TH)—that die [30,31]. Among the TH-positive neurons, it is mostly the neurons with (DA-derived) neuromelanin (NM) pigment that die [30,31], and, among those, predominantly, the neurons with most NM die [32]. NM amounts increase with the concentration of cytosolic DA [33] and, roughly, with a person’s age [34]. Hence, even if the mechanisms are not well understood (see discussions below), DA and NM are positively associated with PD vulnerability.

The SNc can be divided into subregions, of which one method is based on calbindin D28K immunohistochemistry. This method distinguishes between calbindin-rich “matrix” areas and calbindin-poor “nigrosomes”. In the nigrosomes, the neurons have more NM and are more densely packed, and here, the PD-induced neurodegeneration starts earlier and is more severe [35]. As explained below, the calbindin presence in the matrix seems to have a protective effect against PD while having an inhibitory effect on DA production.

### 2.5. The Axonal Arbor Degenerates First

SNc DA neurons have a massive axonal arbor in the striatum (Figure 1 and see below). At both PD motor symptom onset and death, the loss of striatal DA markers exceeds that of SNc DA neurons [36]. This suggests a “dying-back process” (retrograde degeneration) in PD of SNc DA neurons [11,37].

### 2.6. Degeneration of Other Catecholamine (CA) Neurons in the Midbrain

Because PD is believed to be predominantly caused by the degeneration of SNc DA neurons (A9 area), this article will mainly focus on those cells. However, it is essential to realize that, in PD, DA neurons in the SN pars reticulata (SNr), the retrorubral field (A8 area), and the ventral tegmental area (VTA, A10 area) also preferentially die, although to a lesser extent than in the SNc [30,31,38,39]. The LC shows similarities with the SNc in that its catecholaminergic (NE neurons in this case) NM-producing neurons preferentially die [30,40,41], and an effect of this neurodegeneration on the SNc has been proposed [41,42]. In the Braak model of PD staging, LBs are formed in the LC before they are formed in the SNc [19].

**Figure 1 ijms-25-02009-f001:**
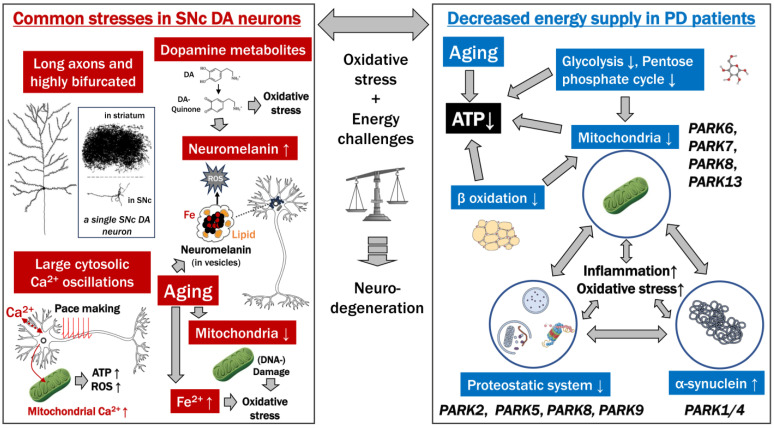
Specific vulnerability of SNc DA neurons: high DA production causes oxidative stress and energy challenges that contribute to Parkinson’s disease. Dopamine neurons have long, highly branched axons. The boxed figure showing the part of a single SNc DA neuron in the SNc and its axonal arbor in the striatum is modified from Matsuda et al. 2009 [43] (with permission from Copyright 2009 Society for Neuroscience). DA-quinone derived from DA is an important source of oxidative stress. DA metabolites are building blocks of the melanin component of neuromelanin (NM), which forms an association within membraned organelles with proteins, lipids, and metals. Aging increases the amount of NM, as does the accumulation of DNA damage in mitochondria and the concentration of iron, which both contribute to oxidative stress. In order to generate a robust pacemaking system with abundant ATP, SNc DA neurons have large Ca^2+^ oscillations that increase the intra-mitochondrial Ca^2+^ concentrations and thereby increase both ATP and ROS production. In PD, the energy metabolism is generally impaired. Dysfunction of the glycolytic system, pentose phosphate pathway, mitochondria, and β-oxidation can commonly cause a decrease in ATP. Glycolysis, the pentose phosphate pathway, and β-oxidation interact with or constitute mitochondrial functions. Proteostatic functions (protein metabolism involving building and degradation) require ATP and directly interact with mitochondrial functions, and in PD and aging, deficiencies in all three aspects are observed. The misfolding of α-synuclein also affects mitochondria and proteostatic competence. Abnormalities in mitochondria, the proteostatic system, and α-synuclein can all lead to neuroinflammation and elevated oxidative stress. Genes responsible for familial PD generally affect one of these factors. Aging is associated with a decrease in ATP. Ultimately, excessive oxidative stress and energy challenges outweigh compensatory mechanisms and can lead to neurodegeneration. Most drawings in this figure are obtained from “Mind the Graph” (https://mindthegraph.com/my-creations/, accessed on 1 October 2023).

### 2.7. The Nucleus Accumbens (NAc)

The axons of DA neurons whose cell bodies reside in the SNc predominantly project to the dorsal striatum, while DA supply to the NAc mainly derives from neurons with their cell bodies in the VTA, a brain region directly adjacent to the SNc [44,45]. However, this specialization is not absolute; VTA DA neurons also innervate the dorsal striatum, and SNc DA neurons also innervate the NAc [44]. Although to a lesser degree and at a later stage than in the SNc, DA neurons in the VTA are affected by PD as well [38,46]. In PD patients with cognitive disorders, the NAc shows atrophy and a statistically significant decrease in volume [47,48], which is thought to be primarily caused by the dying of DA neurons.

### 2.8. Summary

PD does, at least in most cases, not start in SNc DA neurons. However, it results in their preferential degeneration, especially of those that reside in the “nigrosome” regions of the SNc and starting with their axonal arbors in the striatum. CA neurons in brain regions other than the SNc also preferentially die. Among SNc DA neurons, those with higher NM concentrations die, suggesting a critical toxic role of CA production in PD susceptibility. However, there is no evidence that biological differences in CA production are sufficient to induce PD, and CA production rather appears to create a common vulnerability that, in the case of aging and PD, is acted on by other factors. In the majority of PD patients, Lewy bodies/α-synuclein appear to contribute to the initiation and anatomical spread of the disease. However, they do not always lead to PD and are not always found in PD. Thus, critical questions in PD are (1) why are SNc DA neurons especially vulnerable and (2) which factors can act on this vulnerability and thereby induce neurodegeneration?

## 3. Comparison of SNc DA Neurons between Humans and Other Species

### 3.1. An Evolutionary Ancient System

DA is a neurotransmitter that promotes motor activity/control and learning in a wide variety of animals [49,50,51], including the primitive nematode *C. elegans* [52]. Already from the evolutionary level of jawless vertebrates, in the basal ganglia, SNc/VTA DA neurons projecting to the striatum are found, and from the level of amniotes (reptiles/birds/mammals) separate SNc and VTA regions can be distinguished that predominantly project to the dorsal and ventral striatum, respectively [53]. Conserved properties from *C. elegans* to humans also seem to be that DA concentrations decrease with the age of the animals (for *C. elegans*, see [54]) and that DA is a neuromodulator that is importantly released by means of extrasynaptic “volume transmission” [55]. Volume transmission means a diffuse release and signaling which is not restricted to the synaptic clefts and is slower but reaches more targets than the restricted (private) mode called “wiring transmission”. To at least a limited extent, many signaling molecules released in the brain, including glutamate and gamma-aminobutyric acid (GABA), appear to use both pathways, but DA, like some other “neuromodulators”, is different from more typical “neurotransmitters” in that the primary mode of DA communication is extrasynaptic volume transmission [56,57].

### 3.2. In Non-Human Mammals, SNc DA Neurons Preferentially Die upon Aging like in Humans, but Natural PD May Hardly Exist

Upon aging, also noted in non-human primates, the SNc DA neurons preferentially die, as summarized by Stark and Pakkenberg [10], and can reach a 50% decrease [58], which is similar to what can be seen in healthy elderly humans. Aged rhesus monkeys display significant impairments in performing delicate motor tasks, and the clinical rating scale correlates with their loss of TH-positive neurons [58]. Also, in aged mice, the combination of locomotor impairments, loss of DA neurons (although mainly in the VTA), or their degeneration (characterized by fragmented mitochondria), and a reduction in striatal DA levels has been observed [59]. However, even though DA neuron vulnerability upon aging is common among mammals, PD is only common in humans. Only recently was the first case of natural PD found outside humans, namely in a cynomolgus monkey that showed classical signs of PD, which could be ameliorated by treatment with L-DOPA [60]. However, this case may be more similar to familial PD than idiopathic PD in humans, as this monkey had mutations in the *LRRK2* gene [60,61]. Li and co-workers, in 2021, calculated that the incidence rate of PD cases in monkeys and humans may be similar but in monkeys, because of their low numbers in captivity, the chance of detection is minimal [61]. Whether that is correct reasoning can be debated, but for animals that are abundantly kept as pets, like cats and dogs, it appears reasonable to assume that true PD is absent. More animal individuals with PD will likely be found, but the overall data indicate that the incidence rate in humans is much higher than in most, if not all, other species.

### 3.3. Only in Humans Neuromelanin (NM) Has Abundantly Been Found

Another at least relatively unique feature of humans appears to be the abundant NM in SNc DA neurons. However, NM is also found in the SNc of non-human mammals [62,63], and non-human primates may not become old enough or may not have been sufficiently investigated at old age for a valid quantitative comparison. Nevertheless, given that NM is produced upon excess cytosolic DA [64] and that the concentration of NM is positively correlated with the degree of neurodegeneration [34], it is attractive to assume that there is a connection between the seemingly uniquely high frequencies of PD and high NM concentrations in humans. This is corroborated by a study showing that recombinant expression of tyrosinase in the rat and mouse SN, leading to the gradual production of NM-like pigment in a multiple-month period, also leads to PD phenomena [65].

Some studies suggested that humans have higher levels of DA in their striatum than other animals, including apes, but that conclusion seems to be questionable because those studies were only based on analysis of TH transcripts or proteins (the latter by immunohistochemistry) in the striatum [66,67,68,69]; those observed species-specific differences may have been caused by differences in the abundance of non-DA dopaminergic TH^+^ interneurons [70] and species-specific differences in anti-TH antibody reactivities upon immunohistochemistry.

### 3.4. Artificial Animal PD Model Systems

By means of neurotoxins, usually 1-methyl-4-phenyl-1,2,3,6-tetrahydropyridine (MPTP), PD model animals can be created in which the animals show many classical PD signs [71]. However, compared to human PD, these systems tend to lack the slowness of progression and do not replicate the “axonal arbor first” sequence of SNc DA neuron degeneration. In contrast, the group Surmeier established a transgenic mouse system in which gradual loss of mitochondria in SNc DA neurons not only induced PD phenomena but also showed slowness of progression and “axonal arbor first” phenotypes reminiscent of human PD [72]. Interestingly, combined studies imply that sufficient DA production in either the striatum (by release from axons of SNc DA neurons) or SNc (by somatodendritic release from SNc DA neurons) is sufficient for prohibiting Parkinsonism [73], suggesting a critical role of the substantia nigra pars reticulata (SNr) neurons on which both DA pathways are believed to have an overall inhibitory effect [74].

### 3.5. Summary

In summary, the vulnerability of DA neurons appears to be evolutionary ancient, including their preferential degeneration upon animal aging. The reason why natural idiopathic PD has only been observed in humans is unclear, and it may be related to the same unknown reasons that also cause NM to be the most abundant in humans (see a list of arguments in Table 1). Artificial animal model systems, for example, by using neurotoxins, show that animals can exhibit PD symptoms if SNc DA neurons are sufficiently damaged, suggesting that quantitative rather than qualitative factors cause the differences between humans and other species in acquiring natural PD.

## 4. The Bioenergetic Demands of SNc DA Neurons and a Role for Calcium in Their Vulnerability

### 4.1. Shared Features among Neurons Susceptible to PD

By studying the different neuron populations with LB in PD, Braak and co-workers concluded all of them to be projection neurons with extensive axon length and poor axonal myelination among their common factors [20] (Table 2). For example, in PD, the parasympathetic unmyelinated preganglionic fibers are the predominant site of lesions in the vagus nerve. In the olfactory nerve, α-synuclein accumulation is found in the olfactory bulb, olfactory tract, and anterior olfactory nucleus, with the olfactory tract being constructed mainly from unmyelinated axons [76]. Furthermore, in PD, in cardiac sympathetic nerves, unmyelinated axons account for the overwhelming majority (98.2%) of axons with α-synuclein aggregates, and they are lost at a higher frequency [77,78]. In the skin of PD patients, α-synuclein deposits were found in unmyelinated nerve fibers of the autonomic nervous system in association with length-dependent nerve fiber loss [79]. Braak and colleagues [19,20] postulated that poor axonal myelination is a cause of stress because it is associated with a higher requirement of energy for the transmission of impulses [80].

The vulnerability concept by Braak et al. [20] was later modified by Sulzer and Surmeier, 2013 [35], who concluded common factors in different populations of PD-susceptible neurons to be “autonomous activity, broad action potentials, low intrinsic calcium buffering capacity, poorly myelinated long highly branched axons and terminal fields, and use of a monoamine neurotransmitter” (Table 2). Similar to the proposal by Braak et al. 2003 [20], Sulzer and Surmeier [35] postulated that energy demands play a major role in the cell vulnerabilities underlying PD, but instead of the focus on poor myelination, they focused on the energy burdens on the cell body of maintaining a massive axon system and continuous firing (autonomous pacemaking). Surmeier especially strongly expressed PD vulnerability via high bioenergetic demands in a presentation in 2016, in which the type of sustained (tonic) firing by SNc DA neurons (see below) was considered “bioenergetically expensive” and to cause these cells to be “close to a bioenergetic cliff” [84]. Also, in later studies, when discussing PD vulnerability, the Surmeier group kept emphasizing that “SN DA neurons have a high basal bioenergetic demand” [81], which may be true but at least deserves some nuance as the demand may not be higher than in PD-resistant neurons (see below).

Although gradually modifying the original concept of PD vulnerability by Braak and co-workers, in 2013, Sulzer and Surmeier [33] still assumed that SNc DA neurons were “unmyelinated or thinly myelinated” and that this contributed to their vulnerability (Table 2). However, we are not aware of primary literature showing evidence for the postulated poor myelination of SNc DA neurons, and the Surmeier group, while in their later studies continuously improving the knowledge of other stressors of SNc DA neurons, appears to not mention poor myelination of these neurons anymore [85].

Compared to many other neurons, SNc DA neurons are unusual by having unmyelinated varicose “bead” structures within their extensive axonal arbor from which they can release DA [86,87], but this may not have been referred to in the early studies by Braak when mentioning “poor myelination”.

### 4.2. SNc DA Neurons Have a Massive Axonal Arbor in the Striatum Involved in “Volume Transmission” and Tonic Releasing of DA

Compared to some other types of neurons, the number of SN dopaminergic (DA) neurons is relatively small, with only about 200,000 to 420,000 in adult humans [88]. However, these neurons have a large volume as they are extensively branched in the striatum—the volume of the combined axons of a single neuron can be larger than 1 mm^3^—and individually exert influence over a large number of striatal neurons (75,000 on average in rat) while innervating both the striatum striosomes and matrix compartments [43] (Figure 1). Extrapolating these numbers to humans, it was estimated that one single human SNc DA neuron may have 1 to 2.5 million striatal synaptic sites [89]. The fact that SNc DA neurons in humans are over 4 m in total length and have over 1 million synapses presumably leads to high energy costs and makes them vulnerable to energy deprivation [90]. Besides being involved in specific signaling at synapses, DA is also a neuromodulator released by “volume transmission”, in which release is followed by diffusion for widespread activation of many target cells, and—especially given that DA is released by volume transmission from abundant varicose structures within the SNc DA axonal arbor [86,87]—the number of striatal sites affected by a single SNc DA neuron is even higher than the synaptic contacts alone. Overall, the SNc DA system does not seem to be developed for selectively targeting precise circuitries, and this “broadcasting” (spatially nonselective action) effect [91] in the striatum is further enhanced by SNc DA neurons being in communication with each other and tending to be synchronized (fire together) [92,93,94].

By continuous firing via pacemaking activity (see below), SNc DA neurons continuously release a tonic level of DA in the striatum, which is the background against which increases or decreases in striatal DA concentrations are interpreted as validation for learning and starting activities [94,95]. It is believed that besides slow and diffuse (“broadcasting”) signaling by volume transmission, DA from SNc DA neurons can also participate in the stimulation of neural circuits with more spatiotemporal precision [87,96]. Regardless, for understanding the energy demands of SNc DA cells, their massive axonal arborization, in combination with tonic firing and volume transmission release of DA, appears to be the most important.

A positive effect of DA predominantly functioning via volume transmission is that the DA shortages associated with PD can at least partially and temporarily be restored by the non-targeted supply of L-DOPA (levodopa).

### 4.3. Pacemaking Activity

Neurons in the SNc are continuously active in vivo, with very broad spikes that dissipate ionic gradients, especially calcium gradients [33]. Other autonomic neurons, especially those of the enteric nervous system [97], are also spontaneously active and have a wide range of spikes. Compared to DA neurons in the VTA, DA neurons in the SNc maintain striatal DA concentrations by the generation of regular (oscillatory) action potentials via a process that also involves L-type Cav1.3 Ca^2+^ channels, even in the absence of synaptic input and need to pump intracellular Ca^2+^ back into the extracellular space against a huge concentration gradient which requires considerable energy [98]. This Ca^2+^ system does not affect the pacemaker frequency of signal spiking, which was shown to be determined by TRPC3 and NALCN channels [99], but improves pacemaking robustness by providing a “feed-forward stimulation” system that ensures a sufficient level of ATP for spiking [81,100]. For a mechanical discussion of the effect of Ca^2+^ influx on ATP generation, see the paragraph on mitochondrial dysfunction below.

The Surmeier group postulated that this oscillatory Ca^2+^ pumping, at the interface of energy demands and oxidative stress, is an important contributor to PD vulnerability, and stated: “One neuronal trait implicated in PD selective neuronal vulnerability is the engagement of feed-forward stimulation of mitochondrial oxidative phosphorylation (OXPHOS) to meet high bioenergetic demand, leading to sustained oxidant stress and ultimately degeneration” [101].

The presence versus absence of the calcium-binding protein calbindin-D28K appears to confer relative protection to the DA neurons in the SNc matrix compared to the ones in the SNc nigrosome [102,103]. Calbindin-D28K is believed to work as a Ca^2+^ buffer that protects against Ca^2+^ toxicity, but the trade-off appears to be a reduction in activity; namely, when Calbindin-D28K expression was genetically blocked, VTA DA neurons that otherwise express this protein released significantly more DA [104]. Therefore, the fact that Calbindin-D28K is absent in the SNc nigrosomal DA neurons is probably indicative of a higher demand on Ca^2+^-mediated signaling for increasing DA release.

Despite that the SNc DA neurons do not seem to have higher bioenergy requirements than many PD-resistant neurons (see below), there are compelling indications that their PD vulnerability does relate to bioenergy factors such as mitochondria and Ca^2+^ pumping.

## 5. Mitochondrial Dysfunction

### 5.1. Mitochondria in the SNc DA Neuron Cell Bodies; Energy Demands in the SNc DA Neurons Are Not Especially High

Mitochondria are the energy suppliers of the cell, providing the cell’s bulk of ATP via a process called oxidative phosphorylation (OXPHOS). During this process, mitochondria also produce reactive oxygen species (ROS) that have a variety of functions but, in excess, and in case of mitochondrial damage/leakage, can harm the cell [105]. Apart from ATP generation, mitochondria also function in other processes relevant to this review, such as calcium signaling [106] and iron homeostasis [107,108]. One cell has many mitochondria, each with a limited lifespan, and their turnover includes processes like fusion/fission and mitophagy [109,110,111,112,113]. Liang et al., 2007, found that compared to other midbrain neurons, including VTA DA neurons and non-DA neurons in the SN, SNc DA neurons in mice have relatively low mitochondria mass in their somata and dendritic areas [114]. Therefore, because the SN exhibits low glucose utilization compared to many other brain regions [82], and SNc DA neuron action potentials are at a slow rate [115,116], Liang et al. argued [114]: “Such low metabolic demands may predispose these neurons to contain a low mitochondrial mass, which in turn may predispose these neurons to degeneration when mitochondria function is impaired”. Thus, although the concepts by Liang et al., 2007 [114], and of the Surmeier group differ in claiming low versus high bioenergy demands by the cell bodies of SNc DA neurons, they share the viewpoint that PD vulnerability involves mitochondria being stressed.

### 5.2. Mitochondria in the SNc DA Neuron Axonal Arbor in the Striatum

The SNc DA neuron axonal arbor in the striatum has its own mitochondria and is also locally supplied with energy, although it depends on the cell body for nuclear and other functions [117]. In the striatum of PD patients, when the volume of SNc DA neuron axonal arborizations is decreased, their concentration of mitochondria is increased, suggesting an attempt to compensate [118]. Many factors that affect the quality of mitochondria in the cell body also do so in the axonal arbor [117]. The common idea that in PD, the pathology of neurodegeneration in SNc DA neurons starts in their axonal arbor in the striatum and only later reaches their cell bodies in a “dying-back process” [11,37] does not necessarily correspond to the mechanistic route of events (see arguments in Table 3). Namely, malfunctioning in the cell body may have a more immediate impact on the maintenance of the axonal arbor than on the intactness of the cell body itself.

### 5.3. Calcium and Mitochondria

In the SNc, the cytoplasm of SNc DA neurons undergoes continuous oscillations in Ca^2+^ concentration related to the pacemaking that includes Cav1 channel signaling (see above). The increased cytoplasmic Ca^2+^ concentrations also lead to increased Ca^2+^ concentrations within mitochondria, stimulating OXPHOS and ATP production and increasing ROS [100] (Figure 1). Studies showed that the oxidative stress in mitochondria of SNc DA neurons is considerably higher than in VTA DA neurons, which is caused by the Cav1 channel signaling [120,121]. A therapeutically promising line of evidence is that the drug isradipine, which diminishes cytosolic Ca^2+^ oscillations in SNc DA neurons without altering autonomous spiking or expression of Ca^2+^ channels, reduced mitochondrial stress and increased mitochondrial numbers in the SNc DA neurons of mice [121]. However, in clinical trials, isradipine has arguably not been a significant breakthrough yet in treating PD in humans, although some promising observations were made [122,123,124]. It should also be noted that isradipine, conditionally dependent, was found to reduce DA release in the striatum [125].

### 5.4. Familial PD Types Mediated by Mitochondrial Dysfunction

Many, if not all, of the significant types of familial PD are characterized by gene mutations that directly or indirectly, often by creating proteostatic dysfunction, affect mitochondrial functions [126] (Figure 1, Table 4). For example, PINK1 (PARK6) and PARKIN (PRKN or PARK2), associated with hereditary latent PD, play essential roles in the mitophagy process [109,127]. PINK1 and PARKIN mutations disrupt mitochondrial function and lead to the generation of ROS as well as inflammatory responses [128]. Phosphatase and tensin homolog (PTEN)-induced putative kinase 1 (PINK1) is a mitochondrial serine/threonine kinase that targets damaged mitochondria for degradation. Parkin is an E3 ubiquitin ligase that, as one of its functions, recognizes proteins on the outer membrane of damaged mitochondria and can target these mitochondria for destruction. Parkin deficiencies lead to oxidative stress [129], and more than half of the investigated PARKIN PD cases show an absence of LB pathology, which is one of the arguments that LBs are not necessary for inducing PD.

### 5.5. Animal PD Models Based on Mitochondrial Dysfunction

The majority of animal models for PD are based on rather direct targeting of mitochondrial functions, i.e., by inhibition of complex I [132], and include the two examples mentioned earlier in this manuscript. MPTP, which causes PD symptoms in both human and animal models, is metabolized in the brain by glial cells to a mitochondrial toxin, MPP^+^, that is taken up selectively by monoaminergic neurons [133,134]. In the earlier mentioned study by the Surmeier group, a genetic mouse PD model was created in which the nuclear gene for the core subunit NDUFS2 of mitochondrial complex I could be deleted by adenovirus vector injection in the SNc region, creating a slow decrease in the number of mitochondria because of the stability and longevity of NDUSF2 protein [135]; this mouse model provides yet another line of evidence that mitochondrial dysfunction alone can be sufficient for causing SNc DA neuron degeneration and PD symptoms.

## 6. Intracellular Toxicity Directly Related to DA or Its Derivates

### 6.1. Vulnerability of Catecholaminergic Neurons in PD

The vulnerability of catecholaminergic neurons (DA and NE neurons), such as SNc DA neurons and locus coeruleus (LC) NE neurons, combined with the general resistance of GABAergic neurons, suggests that the type of neurotransmitter is an important determinant of neuron susceptibility in PD [136]. However, the type of neurotransmitter is not a necessary determinant, as exemplified by the PD vulnerability of cholinergic, glutamatergic, substance P, GABAergic, and glycinergic neurons in the pedunculopontine nucleus [137,138].

### 6.2. Dopamine (DA)

Dopamine is inherently unstable and can produce reactive oxygen species (ROS) via auto-oxidation with metal ions such as Fe^3+^ as catalysts and generate DA- and DOPA-quinones (Figure 2) [139,140]. Quinones are highly reactive molecules that increase oxidative stress [141]. Except for by autoxidation, cytoplasmic DA can also be discarded enzymatically, involving monoamine oxidase (and catechol-O-methyl transferase), but this process generates reactive H_2_O_2_ (Figure 2). Thus, if synthesized DA is not delivered into vesicles and secreted, this leads to oxidative stress within the cytoplasm.

A standard theory for the incorporation of DA metabolites into neuromelanin (NM) is that this is a safe way to store excess free DA molecules in the cytoplasm and thereby reduce oxidative stress. Sulzer et al. [64] showed in vitro that increasing the intracellular DA concentration (by adding L-DOPA) led to enhancements in NM production and neurodegenerative symptoms, which could be reversed by decreasing the cytosolic DA concentration via enhanced uptake of DA into synaptic vesicles by increasing expression of vesicular monoamine transporter molecule VMAT2. This experiment by Sulzer et al. [64] provided straightforward evidence of the cytotoxicity of either cytoplasmic DA itself or its derivate products (including NM).

### 6.3. The Pigment Neuromelanin (NM)

NM in human SNc DA neurons is a dark black/brown melanin pigment that starts to accumulate from early childhood and is enclosed in membranes where it forms aggregates with other molecules [34,144,145] (Figure 1). NM is formed from the oxidation of DA via DAquinone, followed by interaction with other cellular components such as small thiols (cysteine and glutathione), proteins (also α-synuclein), lipids, metals, etc. As mentioned above, a common theory for the incorporation of DA metabolites into NM is that this is a safe way to discard excess free DA molecules from the cytoplasm and thereby reduce oxidative stress.

NM is so abundant (in humans) that it gave the word “nigra” (Latin for black) to the name “substantia nigra” [146], and in micrographs of SNc DA neurons in elderly people, the average NM area was found to be >40% of the total cell body area [147]. This high NM content should already interfere with normal cellular processes by the presence and steric hindrance alone.

NM accumulates so much iron that it can even be detected by magnetic resonance imaging [148]. Complexation by NM is believed to shield the metal from the cytoplasm and to also lower the one-electron reduction potential of the iron ions, making such ions more difficult to reduce by mild reductants; while this helps to protect against iron-induced oxidative stress under extreme conditions, the iron may be released from NM and become cytotoxic [149,150].

As mentioned above, NM amounts increase with the amount of cytosolic DA [64], with DA being a source for the melanic component of NM [34]. However, the biological function of NM—if it has a biological “function” at all—is not very clear [34], and, depending on conditions, NM can both enhance and reduce neurodegeneration [151]. Critical observations are that (i) those SNc DA neurons that have most NM preferentially die in PD [34] and that (ii) a rodent PD model system based on recombinant expression of tyrosinase in SNc DA neurons shows a coincident development of PD symptoms and NM-like pigment [65]. Thus, either (a) NM as a material, (b) the steric hindrance by NM presence, and/or (c) a molecule of which the abundance correlates with NM abundance increases neurodegeneration if at high concentration. After SNc DA neurons have died, the released NM can enhance local toxicity via the stimulation of microglia and inducing neuroinflammation, as reported by Zecca et al., 2008 [152].

Humans have more NM in their SNc DA neurons than other species (see above), which may represent a higher degree of toxicity and thereby explain why only in humans PD is common, either by the toxicity of NM (-abundance) itself or by the higher concentrations of intracellular DA that give rise to NM. Importantly, it should be realized that NM generation is in the SNc DA neuron cell bodies and not in their axonal arbor in the striatum where DA is produced as well, concluding that the PD vulnerability of humans may have to do with a high demand for local DA production in the SNc.

### 6.4. Summary

The high cytoplasmic DA expression in human SNc DA neuron cell bodies, evidenced by NM formation, leads to enhanced oxidative stress from DA metabolites and probably to enhanced metabolic stress by steric and possibly other effects from abundant NM. This should make it harder for mitochondria to fulfill their tasks, and NM is probably an important contributor to PD development.

## 7. Risk Factors of PD

### 7.1. PD Risk Factors, General

Among the known risk factors other than aging are male sex, routine pesticide exposure, occupational solvent exposure, caffeine nonintake, nonsmoking, type 2 diabetes, lack of exercise, and low plasma uric acid levels in men [153]. Several genetic factors increase the risk of PD, with the ones best-known and conferring the highest risk being named PARK genes (Table 4). Although these genes contribute to a variety of cellular processes, they generally have rather direct effects on mitochondrial or proteostatic functions [126,154] (Figure 1); in addition to the PARK genes, many other potential genetic risk loci have been identified as well [155]. However, it is essential to realize that most PD cases are considered idiopathic, thus primarily determined by factors other than patient-specific genetic susceptibilities.

### 7.2. Exposure to Toxins

Environmental toxins can also cause or contribute to PD development [156]. The most famous case is the above-mentioned agent MPTP (1-methyl-4-phenyl-1,2,3,6-tetrahydropyridine). MTPT was discovered around 1980 when drug users, with a first case in Maryland and then more cases in California, started using 1-methyl-4-phenyl-propionoxypiperidine, a meperidine analog and a new synthetic illicit drug that supposedly had “heroin-like” qualities but contained MPTP as a by-product from synthesis; this MPTP was found to induce PD-like symptoms that could be alleviated by L-DOPA treatment [157,158]. Nowadays, various toxins, including metals and a variety of agricultural pesticides such as rotenone, have been associated with increasing susceptibility to PD [156,159]. As mentioned earlier in this article, many of these toxins are believed to promote PD via their detrimental effect on mitochondria, and the PD type and progression may somewhat differ from other causes. To give one more example, very recent epidemiological studies have reported that exposure to trichloroethylene (used as industrial degreasing solvent), which disrupts mitochondrial complex I, increases the prevalence of PD but produces a clinical form with more frequent resting tremor and less frequent rapid eye movement (REM) sleep behavior disturbances, olfactory loss, constipation, and urinary disturbance [160].

### 7.3. Reduction in the Level of Reduced Glutathione Causes Oxidative Stress

Glutathione is an antioxidant, and a decreased concentration of reduced glutathione in the brain regions indicates oxidative stress. One of the pronounced biochemical changes seen in PD is a reduction in reduced and total glutathione levels in several brain regions including in the SN [161,162,163]. Rather than only indicating oxidative stress, decreases in reduced glutathione levels are also thought to directly enhance PD vulnerability [163].

### 7.4. Increased Iron Levels Cause Oxidative Stress

Iron is also a significant component of oxidative stress and is considered an essential player in the pathogenesis of PD [164]. In PD patients, there is a significant increase of iron in the SN, while not in several other brain regions, suggesting a causative relationship between iron and PD [165,166,167,168]. Ferroptosis is a type of programmed cell death characterized by iron dependency and the accumulation of lipid peroxides, and it is genetically and biochemically distinct from other regulated cell deaths, such as apoptosis [169]. Ferroptosis is initiated by the failure of glutathione-dependent antioxidant defenses, leading to unchecked lipid peroxidation, ultimately resulting in cell death. It has been reported that ferroptosis is also associated with the death of DA neurons in PD [170].

### 7.5. Aging Increases Various Risk Factors

The average age of PD onset in populations that are primarily Caucasian is 58.9 years old [171]. Aging is characterized by a general deterioration of functions, including those predisposing to PD, such as deteriorations of proteostasis and mitochondrial functions [172]. Mitochondrial dysfunction in aging is characterized by an accumulation of genetic damage, increased oxidative stress, and decreases in the number of mitochondria [173,174]. In particular, in the SN of the aging brain—although to lesser extents than in age-matched PD patients—oxidative stress is increased by substantial increases in iron concentration [175] and decreases in the levels of reduced glutathione [176]. Furthermore, the aging brain has an enhanced immune status (low-grade inflammation, also known as “inflammaging”), which additionally interferes with brain homeostasis and puts cells under stress [177]. The fact that neurons, most of which cannot replicate or be replaced, can live for many decades is remarkable in itself, and it appears only logical that under the increased stresses from aging, they will eventually deteriorate. Probably all people, if they would live long enough, would eventually develop PD(-like symptoms).

### 7.6. Is Non-Secretion of DA a Risk Factor, and a Reason Why Regular Smoking May Be Protective against PD?

Interestingly, various mutations are believed to decrease DA release before the cells degenerate [178]. If this occurs without a decrease in DA synthesis, this might lead to increased cytotoxicity by accumulated intracellular DA [64]. Therefore, we have speculated that regular DA release by regular physical exercise may be beneficial against PD [32]. Similarly, we here speculate that it may explain why smokers, who regularly release DA via stimulation by nicotine, are, on average, better protected against PD than nonsmokers [179]. Nonsmoking has even been described as one of the most substantial risk factors [180].

## 8. Energy Status and PD

### 8.1. Metabolic Alterations in PD

With mitochondrial functions having such a prominent role in PD vulnerabilities, it is only logical that various bioenergetic factors may impact PD, including nutritional status. Several questions must be addressed to understand energy deficits in neurodegeneration, such as its role in disease onset and whether restoring energy may halt cell death. Minor energy shortfalls can hamper synapses, while significant deficits can trigger cell death [181].

Early PD is associated with various metabolic abnormalities, such as a decline in mitochondrial β-oxidation and reduced long-chain acylcarnitines [182]. In epidemiological studies, weight loss has been observed in PD patients before disease onset [183], and fat reduction is the primary driver of this pre-onset weight loss [184]. A systematic review of multiple studies found that malnutrition prevalence in PD varies from 0 to 24%, with the malnutrition risk ranging from 3 to 60% [185]. Moreover, progressive weight loss is observed during PD [186] and has been associated with increases in cognitive decline [187] and mortality [188]. In a human brain imaging study in PD, a widespread reduction in cerebral glucose uptake was observed by [18F] FDG PET along with cognitive decline [189]. More specifically, analysis by [18F] FDG PET also indicated, in PD, a reduced glucose uptake in the substantia nigra [83,189].

On the other hand, suggesting that different types of energy unbalances can promote PD, there is a heightened incidence of type 2 diabetes in PD patients [190].

### 8.2. Disruption of the ATP-Producing System in PD

Several lines of evidence support that adenosine triphosphate (ATP), or sufficient levels of ATP (energy), is protective against PD. For example, phosphorus and proton magnetic resonance spectroscopy studies have reported decreased ATP levels in the PD midbrain, putamen, and muscle [191]. Moreover, various mutations causing PD, including in PINK1 and SNCA, have been associated with impaired ATP production [127,192,193]. Furthermore, the research underscores ATP’s protective potential against protein aggregation [194] and α-synuclein toxicity [195].

Also, evidence suggests that energy replenishment may prevent cell death, as maintaining a balance between ATP production and ROS appears to be vital for neuronal survival. For example, interventions enhancing ATP levels have demonstrated therapeutic potential in PD models [196]. An underlying reason may be that PD-associated mitochondrial dysfunction may prompt inefficient ATP production, exacerbating ROS generation and activating aerobic glycolysis (neuronal Warburg effect) [197]. Interestingly, it has been reported that α1-adrenoceptor blockers (terazosin, doxazosin, and alfuzosin), which potentiate glycolysis, appear to help prevent PD-associated cognitive decline [198].

Purine nucleotides are synthesized by two pathways: a “de novo” synthetic pathway that biosynthesizes purine nucleotides using ribose-5-phosphate provided by the pentose phosphate pathway as material and a nucleotide re-synthesis (“salvage”) pathway using bases and nucleosides as materials (Figure 3). Autopsy brain studies have reported that the pentose monophosphate circuit may be extensively impaired in early PD [26]. The salvage synthesis pathway, which uses degraded nucleotides (IMPs) from ATP to produce ATP, is highly developed in humans and is essential for efficiently maintaining ATP [199]. In the salvage circuit, the process by which hypoxanthine, produced in the pathway where uric acid is synthesized from IMP, is recovered as IMP by hypoxanthine phosphoribosyltransferase (HPRT) is important. We have found that augmentation of this salvage synthesis pathway by administering febuxostat and inosine to patients can increase their plasma hypoxanthine levels and may improve the clinical manifestations of PD [200]. Figure 3 summarizes the energy impairments reported in PD.

## 9. Potential Therapies and Questions That Need Addressing

As explained in the introduction of this review, there are no well-established therapies yet that slow down the progression of PD. However, when discussing the underlying vulnerabilities of PD, as in the present article, it appears that there are a few logical therapeutic targets, as summarized in Table 5.

We believe that the best chances for developing therapies to slow down PD progression may be by targeting the processes for which there may be evidence already, namely by reducing intracellular DA accumulations (we speculate that this may explain the protective effect of smoking) and improving the energy status of PD patients.

Given the beauty of the model and the support from studies in rodents and epidemiological data, we also think that the attempts to reduce PD vulnerability by calcium channel blockers like isradipine should be continued. We speculate that this type of drug may also have an adverse effect by blocking DA release and thereby increasing DA toxicity so it would be good to test such drug together with a therapy aimed at enhancing DA release (Table 6).

Table 6 also summarizes other types of research that we think should be performed. Important is to find out why only humans have been found to have idiopathic PD, as such understanding may also help to prevent the disease. Related to that question are the questions of whether humans release more DA into the SN and why.

Overlapping with Table 5, very important research topics, in our opinion, are to find out how regular somatodendritic DA release in the SNc can be stimulated and whether that is protective against PD, and also to test further if improving the energy status of patients helps to protect them against PD and investigate the mechanism of that protection (Table 6).

## 10. Limitations of This Study

A general weakness of many reviews, including this one, is that topics are broadly summarized, and thereby, important information is lost and neglected. For example, rather than trying to summarize PARK genes into broad categories with respect to PD vulnerabilities, it would also be valuable to discuss each of them individually and to not only look for the common aspects.

Some of the topics which deserve more attention than given in the present review are the impact of the immune system on PD, and the dissection of PD per feature, per brain region, and per stage of progression. Also, whereas our review focuses on the initial causes/vulnerabilities, the actual processes of neurodegeneration are an exciting research field in itself.

In PD, it is well known that as the disease progresses, lesions spread beyond the substantia nigra, including becoming widespread in the cerebral cortex. Therefore, a critical question concerns whether and how the pathophysiology of the neuronal cells of the substantia nigra, the topic to which our review was mostly restricted, affects other brain regions.

The biggest weakness of the present study, however, which is inherent to the fact that arguably good rodent models of PD are only beginning to be developed, is the lack of good quantitative data, so that many issues remain speculative.

## 11. Conclusions

This review discusses two main underlying causes of SNc DA neurons’ specific vulnerability: (i) their high bioenergetic demands and (ii) their high DA production. We summarized our discussions in Figure 1.

To a large degree, as shown by animals naturally showing similar phenomena, the preferential dying of SNc DA neurons upon aging seems unavoidable. In humans, however, the stresses on these neurons seem to be exceptionally high, as only humans have abundant NM and idiopathic PD. Although the first degeneration symptoms of SNc DA neurons in PD are believed to start in the axonal arbor in the striatum, there is the possibility that the cause of PD starts in the cell bodies in the SNc. This is the location where two main stressors, namely NM formation and cytosolic Ca^2+^ oscillations, occur.

Although we are not aware of studies that addressed this, we speculate that in human SNc, the local somatodendritic DA release may be exceptionally high compared to other species, possibly to enhance the within-SN pathway for DA from SNc to SNr.

The DA production system of SNc DA neurons seems to be tailored toward high DA production, such as by involving cytosolic Ca^2+^ oscillations without the protective presence of Calbindin-D28K, thereby creating a lot of oxidative stress for the mitochondria. That the mitochondria of these neurons are already stressed under non-PD conditions appears to be proven by the fact that various factors that damage mitochondrial functions seem to tip them over the edge and enhance PD. Mitochondria are the energy providers of the cell, so it is only logical that energy factors like nutrition also affect PD.

Logical approaches against PD progression include (i) reducing intracellular DA accumulation, (ii) blocking cytosolic Ca^2+^ oscillations, and (iii) providing bioenergetic support to PD patients.

As for reducing intracellular DA accumulation, especially in the SNc DA neuron cell bodies, we are not aware of any current treatments aimed at this. However, we assume that the regular DA release may explain the lesser PD risk of smokers. We also have speculated that regular exercise may increase protection against PD by releasing DA [32]. It is puzzling why isradipine, which blocks the cytosolic Ca^2+^ oscillations, has not been more successful against PD yet. We wonder if the benefit of protection against Ca^2+^ influx may not be masked by a decrease in DA release, possibly leading to increased DA-derived cytotoxicity. Although it is not yet determined whether nutrition and other bioenergetic factors can be the primary drivers of PD or not, they include risk factors, and it has been shown that supporting the bioenergetics of PD patients is beneficial to them.

## Figures and Tables

**Figure 2 ijms-25-02009-f002:**
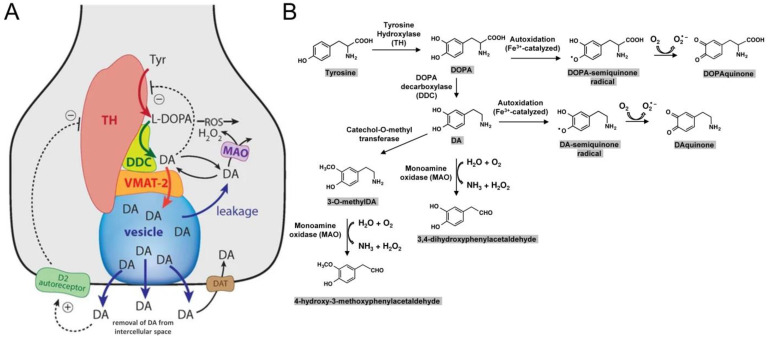
Overview of the synthesis and regulation of DA, which are similar in both the somatodendritic and axonal parts of SNc DA neurons. (**A**) This figure and its legend were adapted with permission from Kleppe et al. 2021 [142]. There is evidence that DA is synthesized from tyrosine (Tyr) by a channeling mechanism with the enzymes tyrosine hydroxylase (TH), dopa decarboxylase (DDC), and the transporter VMAT2 forming a complex at the vesicular membrane. The synthesized DA is stored in synaptic vesicles for release. Significant leakage of transmitter molecules out from vesicles has been observed. Upon arrival of an action potential, vesicles are emptied, and DA is released into the intercellular space. There, DA diffuses to DA-responsive target sites or is taken up by DA transporters (DAT). TH is inhibited by DA, and there is also an inhibition of TH from extracellular DA via D2 autoreceptor signaling. Monoamine oxidase (MAO) forms hydrogen peroxide (a reactive oxygen species (ROS)) during the metabolization of DA, and also autoxidation of DA leads to the production of ROS. (**B**) This figure was inspired by Segura-Aguilar et al., 2014 [143] and shows the biochemistry of some, although not all, of the ROS and quinone-generating reactions that DA(-synthesis) can be involved in.

**Figure 3 ijms-25-02009-f003:**
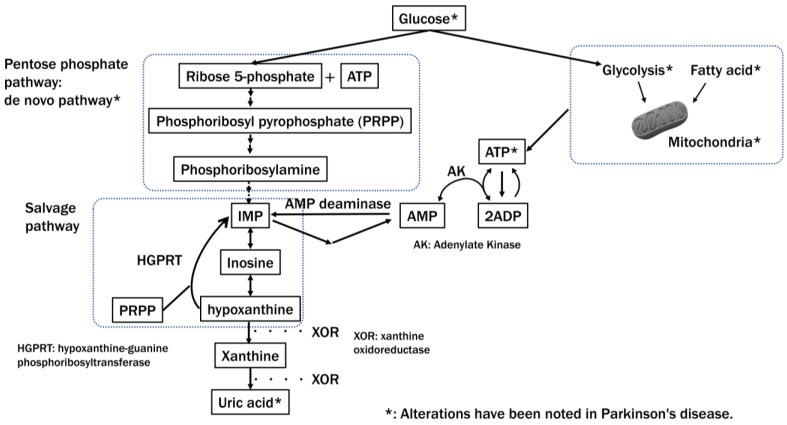
Major ATP synthesis pathways and their impairment in Parkinson’s disease. In Parkinson’s disease, extensive impairment of pathways that synthesize ATP, such as the pentose phosphate pathway, which plays a vital role in producing NADPH and purine synthesis, the mitochondrial and glycolytic systems, and fatty acids, has been reported. Decreased levels of uric acid, a purine metabolite, have also been reported. Figure modified from, and reprinted with permission of, Johnson et al. [199].

**Table 1 ijms-25-02009-t001:** Speculations as to why only humans have been found susceptible to idiopathic Parkinson’s disease.

**Relatively certain contributing factors** Humans become older than most animal species. Humans can survive despite loss of fitness, and wild animals cannot, so only in humans can the disease fully develop. There are many humans. Only elderly humans have abundant neuromelanin (NM) in their SNc DA neurons, and it is known that NM-rich cells predominantly die (reviewed in Nagatsu, T.; et al. *J. Neural Transm.* 2023, 130, 611) [32], and the artificial generation of NM in rodents causes PD-like symptoms (Carballo-Carbajal, I.; et al. *Nat. Commun.* 2019, 10, 973) [65].
**Speculative contributing factors** The fact that human SNc DA neurons have more NM suggests that they somatodendritically secrete more DA in the SN than found in other animals; possibly, compared to other animals, their immediate SNc-to-SNr DA pathway may have a higher importance (speculated in the present study). It has been speculated that humans produce more DA in the striatum than other animals, but this seems only to be based on estimations of tyrosine hydroxylase (TH) amounts that did not consider detection-relevant TH molecular differences between species (e.g., Raghanti, M.A.; et al. *J. Comp. Neurol.* 2016, 524, 2117–2129) [66], while it should also be considered that TH amounts do not directly represent DA amounts (Xenias, H.S.; et al. *J. Neurosci.* 2015, 35, 6584–6599) [70]. The substantial expansion of the telencephalon, especially the neocortex, in humans imposes a considerable burden on subcortical circuits interacting with the telencephalon (*Diederich, N.J.*; et al. *Mov. Disord.* 2019, 34, 453–459) [75].

**Table 2 ijms-25-02009-t002:** Examples of models explaining neural vulnerabilities in PD and some critical considerations.

Examples of Models Explaining Neural Vulnerabilities in PD	
Reference	Statements Relevant to the Model
Braak, H.; Rüb, U.; Gai, W.P.; Del Tredici, K. Idiopathic Parkinson’s Disease: Possible Routes by Which Vulnerable Neuronal Types May Be Subject to Neuroinvasion by an Unknown Pathogen. *J. Neural Transm*. 2003, 110, 517–536 [20].	*[about the common properties of neurons vulnerable to PD]* “All of the vulnerable cells belong to the class of projection neurons. Within this class, only some neuronal types, namely those which generate axons that are disproportionately long in relation to their somata, demonstrate a pronounced tendency to develop the lesions”. “An additional feature shared by all of the endangered neuronal types is that their long and thin axons are unmyelinated or only partially myelinated”. *[about the energy usage by these neurons]* “The maintenance of unmyelinated or incompletely myelinated axons requires prodigious expenditures of energy”.
Sulzer, D.; Surmeier, D.J. Neuronal Vulnerability, Pathogenesis, and Parkinson’s Disease. *Mov. Disord*. 2013, 28, 715–724 [35].	*[about the (common) properties of neurons vulnerable to PD]* “There appears to be a small number of risk factors contributing to vulnerability. These include autonomous activity, broad action potentials, low intrinsic calcium buffering capacity, poorly myelinated long highly branched axons and terminal fields, and use of a monoamine neurotransmitter, often with the catecholamine-derived neuromelanin pigment”. “SNc, LC, RN, PPN, and NBM neurons all have unusually long highly branched axons that are unmyelinated or thinly myelinated”. *[about the energy usage and metabolic stress of SNc DA neurons related to their calcium pumping]* “Calcium entry is energetically expensive because it must be pumped out of the cell against a much steeper electrochemical gradient than any of the other ions”. “One of the ion channels contributing to the basal metabolic stress in SNc DA neurons is the L-type calcium channel”.
Zampese, E.; Surmeier, D.J. Calcium, Bioenergetics, and Parkinson’s Disease. *Cells* 2020, 9, 2045 [81].	*[about the properties of neurons, especially SNc DA neurons, that make them vulnerable to PD]* “The available data indicates that an extensive axonal branching, autonomous pacemaking, and Cav1 channel-mediated feedforward control of mitochondrial OXPHOS (and the consequent mitochondrial oxidant stress) might be key features determining neuronal vulnerability in PD”. “Unlike most neurons, SN DAergic neurons appear to have a high basal bioenergetic demand. This demand may have its roots in several factors. The most important of these is likely to be the neuron’s massive axonal arbor. This arbor creates an anabolic demand, as it has to be supplied with release-related proteins and lipids largely delivered by axonal transport from the somatic region”.
**Some Critical Considerations**	
1. There may not be good evidence that SNc DA neurons are poorly myelinated.	
2. The glucose consumption of the SNc is not especially high (Sokoloff, L.; et al. *J. Neurochem.* 1977, 28, 897–916 [82]; Schröter, N.; et al. *NPJ Parkinson’s Dis.* 2022, 8, 123 [83]), arguing against an explanation of stress in SNc DA neurons due to unusually high energy demands.	

**Table 3 ijms-25-02009-t003:** Where does the degeneration of SNc DA neurons start, in their cell bodies in the SNc or in their axonal arbor in the striatum? (*we consider it most likely that the critical toxicity is generated in the cell body but that its first dramatic degenerative effects appear in the axonal arbor*).

**Arguments that neurodegeneration starts in the cell bodies** Neuromelanin in the cell bodies of SNc DA neurons is quantitatively associated with the degeneration of these neurons (reviewed in Nagatsu, T.; et al., *J. Neural Transm.* 2023, 130, 611) [32]. In a mouse PD model system, the drug isradipine, which reduces Ca^2+^ influx into the SNc DA neuron cell bodies, has a protective effect on DA neurons (Ilijic, E.; et al., *Neurobiol Dis.* 2011, 43, 364–371) [119]. Considering the major direction of the flow of materials and signals in neurons, a disease purely generated in the axonal arbor would not necessarily have a major impact on the cell bodies, whereas vice versa, an impact seems unavoidable; the fact that PD affects both locations supports that the toxic effects start in the cell bodies.
**Arguments that neurodegeneration starts in the axonal arbor** In PD, the degrees of losses of DA and tyrosine hydroxylase activity in the striatum exceed those in the substantia nigra (Hornykiewicz, O. *Neurology* 1998, 51, S2–S9) [37]. α-synuclein pathology is abundant in axons and presynaptic terminals (reviewed by Cheng, H.-C.; et al., *Ann. Neurol.* 2010, 67, 715–725) [36].

**Table 4 ijms-25-02009-t004:** PARK-designated genes.

PARK	Gene	Protein	Function
PARK1, PARK4	*SNCA*	α-synuclein	Uncertain, but misfolding causes Lewy Bodies
PARK2	*PRKN*	Parkin, E3 ubiquitin ligase	Mitochondrial
PARK5	*UCHL1*	Ubiquitin C-terminal hydrolase L1	Ubiquitin-proteasome
PARK6	*PINK1*	PTEN-induced putative kinase 1	Mitochondrial
PARK7	*DJ-1*	Parkinsonism-associated deglycase	Mitochondrial
PARK8	*LRRK2*	Leucine-rich repeat kinase 2	Lysosomal, mitochondrial, microtubule
PARK9	*ATP13A2*	Cation-transporting ATPase 13A2	Lysosomal
PARK11	*GIGYF2*	GRB10 interacting GYF protein 2	Uncertain
PARK13	*HTRA2*	HtrA serine peptidase 2	Mitochondrial
PARK14	*PLA2G6*	Calcium-independent phospholipase A2 enzyme	Cell membrane
PARK15	*FBX07*	F-box protein 7	Mitochondrial
PARK17	*VPS35*	Vacuolar protein sorting-associated protein 35	Retromer and endosomal trafficking
PARK18	*EIF4G1*	Eukaryotic translation initiation factor 4 gamma 1	Transcription
PARK19	*DNAJC6*	HSP40 Auxilin	Synaptic vesicle formation and trafficking
PARK20	*SYNJ1*	Synaptojanin 1	Synaptic vesicle formation and trafficking
PARK21	*DNAJC13*	Receptor-mediated endocytosis 8 (RME-8)	Synaptic vesicle formation and trafficking
PARK23	*VPS13C*	Vacuolar protein sorting-associated protein 13C	Mitochondrial

This table is a summary of similar tables in Kouli et al., 2018 [130] and Day and Mullin 2021 [131]. For PARK3, PARK10, PARK12, PARK16, and PARK22, the genes have not been identified yet, and they are not included in this table.

**Table 5 ijms-25-02009-t005:** Potential therapeutic strategies to reduce stress in SNc DA neuron cell bodies. (*a selected set of possible options; for a broader discussion of anti-PD therapies, see Chopade, P.; et al., Bioeng. Transl. Med. 2022, 8, e10367 [201]*).

**Reducing the accumulation of intracellular DA** (*speculation only*) Regular exercise/sports will probably lead to a decrease in intracellular DA by stimulating DA release (speculated in Nagatsu, T.; et al., *J. Neural Transm.* 2023, 130, 611) [32]. (*speculation only*) Smoking may be protective against PD by the regular release of DA (speculated in the present study), and this effect should be replicated in PD patients (*although preferably not by smoking itself because of its other toxic and carcinogenic effects*). Amantadine is used as an anti-PD drug, and it enhances DA release, although the mechanism of action is not well understood (reviewed by Rascol, O.; et al., *Lancet Neurol.* 2021, 20, 1048–1056) [202].
**Reducing the calcium ion influx** (*not well proven to help against PD in clinical trials yet*) Isradipine is a drug that diminishes cytosolic Ca^2+^ oscillations in SNc DA neurons (Guzman, J.N.; et al., *J. Clin. Invest.* 2018, 128, 2266–2280) [121]. A meta-analysis study suggested that either the use of a dihydropyridine type (like isradipine) or a non-dihydropyridine type calcium channel blocker (CCB) reduces the risk of developing PD (Gudala, K.; et al., *Int. J. Chronic Dis.* 2015, 697404) [203].
**Assuring a proper energy supply** Enhancing glycolysis attenuates PD progression in models and clinical databases (Cai, R.; et al., *J. Clin. Invest.* 2019, 129, 4539–4549) [204]. Data suggest that patients using the glycolysis-enhancing drugs terazosin/doxazosin/alfuzosin are at lower hazard of developing PD compared with users of tamsulosin (Simmering, J.E.; et al., *Mov. Disord.* 2022, 37, 2210) [205]. Administration of febuxostat and inosine resulted in increased hypoxanthine and ATP levels (Kamatani, N.; et al., Gout And Nucleic Acid Metab. 2017, 41, 171–181) [206] and improved PD motor score (Watanabe, H.; et al., *Medicine (Baltimore)* 2020, 99, e21576) [200]. Medicare database analysis suggested that xanthine dehydrogenase/oxidase blockade, associated with enhancing the salvage pathway of ATP, reduces the risk of PD (Song, Y.; et al., *PLoS One* 2023, 18, e0285011) [207].
**Antioxidant** Theoretically, various ways to reduce oxidative stress, including, for example, iron chelation, should have a protective effect on PD. However, so far, these approaches have not been very successful (reviewed by García-Beltrán, O.; et al., *Antioxidants (Basel)* 2023, 12, 214) [208].
**Inhibition of inflammation** In rodents, the drug exenatide is believed to protect DA neurons through inhibition of microglial activation (Kim S.; et al., *J. Endocrinol.* 2009, 202, 431–439) [209]. However, so far, in clinical trials against PD, it probably did not have a significant protective effect (Athauda D.; et al., *Lancet* 2017, 390, 1664–1675) [210].
**Reducing α-synuclein aggregation** There are several therapeutic approaches for reducing α-synuclein aggregation, which showed promise in animal PD models, but none of which has been established as significantly protective against PD in clinical trials yet: for example, NPT200-11 (Price D.L.; et al., *NPJ Parkinsons Dis.* 2023, 9, 114) [211] or Prasinezumab (Pagano, G.; et al., *N. Engl. J. Med.* 2022, 387, 421–432) [212].

**Table 6 ijms-25-02009-t006:** Questions related to SNc DA neuron vulnerability in PD that still need addressing. (*this is only a selected set of interesting questions that have our particular interest*).

**Why is PD only prevalent in humans?** Do in humans the SNc DA neurons somatodendritically release more DA than in other animals? What is the function of DA that is somatodendritically released in the SNc? Is in humans the immediate SNc-to-SNr DA pathway more important than in other animals?
**Does increased DA release help protect against PD?** Does increased DA release, such as by smoking or exercise, help protect against PD? (note: We speculate that the lack of major success in clinical trials that administered nicotine to halt PD progression (reviewed by Quik M.; et al., *Biochem. Pharmacol.* 2009, 8, 677–685 [213]) may not have sufficiently replicated the DA release regimen of a heavy smoker)
**Why has isradipine not been (very) successful against PD?** We speculate that isradipine treatment may counteract its protective effect by also creating a toxic effect, namely by causing a reduction in DA release that may lead to an accumulation of intracellular DA. Therefore, it would be interesting to investigate the therapeutic value of isradipine in combination with a strategy aimed at increasing somatodendritic DA release.
**Can the energy status of SNc DA neurons be improved/protected?** Real-world data demonstrated that ATP-enhancing drugs, which support glycolysis and salvage pathways, could improve PD symptoms or protect against PD progression. The precise mechanisms need to be elucidated, including on whether ATP-enhancing drugs support mitochondrial function in PD.

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
