# Peer review of "Parkinson’s Disease: Cells Succumbing to Lifelong Dopamine-Related Oxidative Stress and Other Bioenergetic Challenges"

_ijms, 2024, doi:10.3390/ijms25042009_

Round 1

Reviewer 1 Report

Comments and Suggestions for Authors

The present manuscript analyses the relevant literature of the last 3 decades dealing with the loss of dopaminergic neurons in Parkinson's disease and ageing. The main focus of this review is on the importance of disturbances in cellular energy metabolism in the loss of dopaminergic neurones. The role of alpha-synuclein, mutations, neuromelanin, neuronal activity and also dopamine itself in the development of cellular energy deficiency states of dopaminergic neurones was discussed, which may ultimately be the cause of the cell death of these neurones. In particular, the paper also focusses on the importance of Ca2+ oscillation and the energy expenditure required for this and the role of mitochondria in this process. A temporary or permanent lack of ATP is discussed with extensive literature citations as the cause of the death of dopaminergic neurones in Parkinson's disease.

This manuscript is well written and clearly discussed.

Minor :

The introduction only mentions L-DOPA therapy as a treatment for PD. Here, therapy with MAO-B inhibitors, COMT inhibitors, deep brain stimulation and experimental cell replacement therapy could also be mentioned. The latter, in particular, because Lewy bodies have also been found in post-mortem cell transplants; this should also be discussed together with the role of alpha-synuclein.

Author Response

We thank the Reviewer for the time and effort to carefully check our paper and the very nice compliments!

Below, we address each of the individual comments by the Reviewer (in Italic font). Changes that we made to the manuscript are highlighted in its “marked” version.

Comments and Suggestions for Authors

The present manuscript analyses the relevant literature of the last 3 decades dealing with the loss of dopaminergic neurons in Parkinson's disease and ageing. The main focus of this review is on the importance of disturbances in cellular energy metabolism in the loss of dopaminergic neurones. The role of alpha-synuclein, mutations, neuromelanin, neuronal activity and also dopamine itself in the development of cellular energy deficiency states of dopaminergic neurones was discussed, which may ultimately be the cause of the cell death of these neurones. In particular, the paper also focusses on the importance of Ca2+ oscillation and the energy expenditure required for this and the role of mitochondria in this process. A temporary or permanent lack of ATP is discussed with extensive literature citations as the cause of the death of dopaminergic neurones in Parkinson's disease.

This manuscript is well written and clearly discussed.

Our response:

We are very pleased with this compliment, especially as we have dedicated a lot of effort to these matters.

Minor :

The introduction only mentions L-DOPA therapy as a treatment for PD. Here, therapy with MAO-B inhibitors, COMT inhibitors, deep brain stimulation and experimental cell replacement therapy could also be mentioned. The latter, in particular, because Lewy bodies have also been found in post-mortem cell transplants; this should also be discussed together with the role of alpha-synuclein.

Our response:

As requested, we have now added this information. On page 1, line 36, it now says:

“Monoamine oxidase type B (MAO-B) inhibitors, catechol-o-methyl-transferase (COMT) inhibitors, and deep brain stimulation therapy are also widely being used, but these treatments cannot stop the progression of the disease either [3]. Even in cell transplantation therapy, Lewy bodies have also been reported to appear in grafts, and the value of the therapy remains to be seen [4–6].”

Again, we like to thank the Reviewer for the support and the improvement of our study.

Sincerely,

Also on behalf of Professor Nagatsu and Dr. Dijkstra,

 Hirohisa Watanabe

Hirohisa Watanabe, MD, PhD

Professor, Department of Neurology, Fujita Health University, School of Medicine,

1-98, Dengakugakubo, Kutsukake-cho, Toyoake, Aichi 470-1192, Japan

E-mail: hirohisa.watanabe@fujita-hu.ac.jp

Reviewer 2 Report

Comments and Suggestions for Authors

In this manuscript, Watabale et al. endeavor to address various facets of the vulnerability observed in substantia nigra pars compacta (SNc) dopamine neurons in Parkinson’s disease (PD). The authors posit that the deleterious impact of dopamine (DA) and its byproducts, coupled with heightened bioenergetic demands, collectively contribute to this susceptibility. Although the paper provides a comprehensive exploration of PD neurodegeneration, the strength of the argument could be enhanced with additional supporting evidence.

Regarding the section on the progression of PD pathology, certain subsections need to be succinctly summarized, while others warrant expansion for a more profound comprehension. In the subsection exploring cellular and regional differences within the SNc, the authors should elaborate on the significance of these variances in the context of PD. For the initial degeneration of the axonal arbor, potential causes and potential regional disparities should be explored. Additionally, it is crucial to investigate how these degenerations align with different stages of PD. Subsections like 'degeneration of other catecholamine neurons in the midbrain' and 'the nucleus accumbens' should be condensed to maintain focus on the main ideas.

In the section comparing SNc DA neurons between humans and other species, the authors have provided an overview of species differences but require additional information to bolster their conclusions. Are there functional disparities in the SNc DA neurons among species? What about variations in the number of SNc DA neurons?

In the third section covering the biogenetic demands of SNc DA neurons, the authors initially address shared susceptibility and vulnerability, followed by discussions on SNc DA volume transmission and tonic release. While these details are valuable, they seem somewhat disjointed. It may be beneficial to briefly explore what makes volume transmission and pacemaking firing of DA unique and how these characteristics may contribute to the susceptibility and vulnerability of SNc DA neurons in PD. This connection would enhance the coherence of the section.

The authors propose certain strategies to mitigate PD neurodegeneration; however, a more robust foundation is needed to substantiate these suggestions. It would be valuable to examine existing literature pertaining to the effectiveness of these approaches in reducing specific aspects of degeneration. Furthermore, an exploration of the outcomes of combining these proposed approaches could provide a more comprehensive understanding of their potential impact on PD progression. 

Comments on the Quality of English Language

NA

Author Response

We thank the Reviewer for the time and effort to carefully check our paper and the insightful comments.

Although the Reviewer considers our work well organized and comprehensively described, he/she suggests deepening the discussion on multiple aspects while descriptions of other topics should be shortened.

As for the choice of the discussed topics and their depth, we believe that in part this is a matter of taste, as not all the reviewers shared the same criticism. However, we now have added an extra figure (Fig. 2) and several new tables (Tables 1, 3, 5, and 6), which highlight some new points and also make the main discussions in this article easier to follow. We also added new chapters, near the end of the manuscript, on research that still needs to be done and the limitations of our article.

We are aware that there are many sub-topics in this manuscript that each deserve much more attention than we can give them. That this has not been done is for space reasons, and sometimes for our lack of the relevant expertise, but mostly because clear information that most definitely needs to be presented is often very scarce. We like to point out that this manuscript has now been reviewed by three expert reviewers, and none of them has made a specific request (apart from more general requests) about study findings that need to be added. We agree that we have not been perfect, but point out that a full discussion of any of the sub-topics would already warrant a full review in itself.

Below, we address each of the comments by the Reviewer (in Italic font). Changes that we made to the manuscript are highlighted in its “marked” version.

Comments and Suggestions for Authors

In this manuscript, Watabale et al. endeavor to address various facets of the vulnerability observed in substantia nigra pars compacta (SNc) dopamine neurons in Parkinson’s disease (PD). The authors posit that the deleterious impact of dopamine (DA) and its byproducts, coupled with heightened bioenergetic demands, collectively contribute to this susceptibility. Although the paper provides a comprehensive exploration of PD neurodegeneration, the strength of the argument could be enhanced with additional supporting evidence.

Our response:

We thank the Reviewer for the compliment that our review provides a comprehensive exploration.

Regarding the section on the progression of PD pathology, certain subsections need to be succinctly summarized, while others warrant expansion for a more profound comprehension.

Our response:

As discussed above, we think that for a large part this is a matter of taste, and in many cases would like to maintain the level of focus. However, we now delve a bit deeper into several topics and try to highlight them better, mostly by providing an extra figure and four extra tables.

In the subsection exploring cellular and regional differences within the SNc, the authors should elaborate on the significance of these variances in the context of PD.

Our response:

The Reviewer is correct that in this text section, this can already be mentioned. Therefore, we have now added, for at the cellular level:

 (page 3, line 121)

“Hence, even if the mechanisms are not well understood (see discussions below), DA and NM are positively associated with PD vulnerability. “  

and for at the SNc regional level (page 3, line 128)

“As explained below, the calbindin presence in the matrix seems to have a protective effect against PD while having an inhibitory effect on DA production.”

For the initial degeneration of the axonal arbor, potential causes and potential regional disparities should be explored.

Our response:

We consider this to be largely unknown, with the potential causes being a main topic of our review. However, related to the Reviewer’s question, we have now added Table 3 (on page 11) titled “Where does the degeneration of SNc DA neurons start, in their cell bodies in the SNc or in their axonal arbor in the striatum?”

Furthermore, in a new chapter titled “Limitations of this study,” we now write (Page 21, line 696):

“Some of the topics which deserve more attention than given in the present review are the impact of the immune system on PD, and the dissection of PD per feature, per brain region, and per stage of progression. Also, whereas our review focuses on the initial causes/vulnerabilities, the actual mechanisms of neurodegeneration are an exciting research field in itself.”

Additionally, it is crucial to investigate how these degenerations align with different stages of PD.

Our response:

See our above response. Furthermore, we have now also included in the described limitations (page 21, line 701):

“In PD, it is well-known that as the disease progresses, lesions spread beyond the substantia nigra, including becoming widespread in the cerebral cortex. Therefore, a critical question concerns whether and how the pathophysiology of the neuronal cells of the substantia nigra, the topic to which our review was mostly restricted, affects other brain regions.”

Although the Reviewer is of course right in that the interregional interactions are an important aspect of PD. However, we do not think that there is a lot of valuable yet simple information that can easily be added to our review. For example, initially, we wanted to also include descriptions of functional changes in the striatum after long PD (after long levodopa therapy), but found this made the manuscript overly complex and unreadable.

Subsections like 'degeneration of other catecholamine neurons in the midbrain' and 'the nucleus accumbens' should be condensed to maintain focus on the main ideas.

Our response:

We understand the argument, but consider it to mostly be about taste, as these are also important questions that are closely related to the main question of our review. Although to a lesser extent, the nucleus accumbens is also innervated by DA neurons from the SNc, and therefore deserves some attention.

In the section comparing SNc DA neurons between humans and other species, the authors have provided an overview of species differences but require additional information to bolster their conclusions. Are there functional disparities in the SNc DA neurons among species? What about variations in the number of SNc DA neurons?

Our response:

We have no added a new Table 1 (page 7), titled “Speculations as to why only humans have been found susceptible to idiopathic Parkinson’s disease.”

In that table, we also speculate on a human-specific contribution to PD-vulnerability by “The substantial expansion of the telencephalon, especially the neocortex, in humans imposes a considerable burden on subcortical circuits interacting with the telencephalon (Diederich, N.J.; et al. Mov. Disord. 2019, 34, 453-459) [75].”

However, overall, we found there to be a frustrating lack of quantitative information that would allow a meaningful answer to the reviewer’s question. Therefore, on page 21, line 708, we now refer to “the lack of good quantitative data so that many issues remain speculative.”

According to reports, humans have about 20x more SNc DA neurons than rats. However, whether to interpret that as more (in absolute numbers) or less (per body weight) is quite complicated, and more detailed information on the systems would be necessary before a meaningful discussion can be had.

In the third section covering the biogenetic demands of SNc DA neurons, the authors initially address shared susceptibility and vulnerability, followed by discussions on SNc DA volume transmission and tonic release. While these details are valuable, they seem somewhat disjointed. It may be beneficial to briefly explore what makes volume transmission and pacemaking firing of DA unique and how these characteristics may contribute to the susceptibility and vulnerability of SNc DA neurons in PD. This connection would enhance the coherence of the section.

Our response:

We have now added some more explanation on volume transmission (page 5, line 201):

“Volume transmission means a diffuse release and signaling which is not restricted to the synaptic clefts, and is slower but reaches more targets than the restricted (private) mode called “wiring transmission.” To at least a limited extent many signaling molecules released in the brain, including even glutamate and gamma aminobutyric acid (GABA), appear to use both pathways, but DA, like some other “neuromodulators,” is different from more typical “neurotransmitters” in that the primary mode of DA communication is extrasynaptic volume transmission [56,57].”

As for the relevance of volume transmission for PD, we already wrote, implying a role in energy demands (page 10, line 347 ):

“Regardless, for understanding the energy demands of SNc DA cells, their massive axonal arborization, in combination with tonic firing and volume transmission release of DA, appears to be the most important.”

Now, we have also added, as a positive aspect (page 10, line 350):

“A positive effect of DA predominantly functioning through volume transmission is that the DA shortages associated with PD can at least partially and temporarily be restored by the non-targeted supply of L-DOPA (levodopa).”

The authors propose certain strategies to mitigate PD neurodegeneration; however, a more robust foundation is needed to substantiate these suggestions. It would be valuable to examine existing literature pertaining to the effectiveness of these approaches in reducing specific aspects of degeneration. Furthermore, an exploration of the outcomes of combining these proposed approaches could provide a more comprehensive understanding of their potential impact on PD progression.

Our response:

We have now added (on page 18) a new chapter “Potential therapies and questions that need addressing” with a new Table 5, titled “Potential therapeutic strategies to reduce stress in SNc DA neuron cell bodies” and new Table 6, titled “Questions related to SNc DA neuron vulnerability in PD that still need addressing. In here, we explore methods that have been tried or that we believe should be tried, to reduce neurodegeneration in PD. As for combining approaches, we believe, as explained in this new chapter and shown in Table 6, that calcium channel blockers like isradipine should be tried in combination with a method that enhances DA release.

We like to thank the Reviewer again for the careful efforts and insightful comments.

Although in part we disagree with the Reviewer on the structure (depth per sub-topic) of our review, the comments by the Reviewer have helped us to make various important improvements. We are very grateful for that.

We hope that the Reviewer can accept our choices of how to organize this review. However, if the Reviewer has very specific additional requests of which facts or studies should be integrated into our review, we will of course try that.

Sincerely,

Also on behalf of Professor Nagatsu and Dr. Dijkstra,

 Hirohisa Watanabe

Hirohisa Watanabe, MD, PhD

Professor, Department of Neurology, Fujita Health University, School of Medicine,

1-98, Dengakugakubo, Kutsukake-cho, Toyoake, Aichi 470-1192, Japan

E-mail: hirohisa.watanabe@fujita-hu.ac.jp

Reviewer 3 Report

Comments and Suggestions for Authors

The manuscript entitled "Parkinson’s Disease: Cells Succumbing to Lifelong Dopamine related Oxidative Stress and Other Bioenergetic Challenges” postulates that the susceptibility of substantia nigra pars compacta (SNc) dopamine (DA) neurons to diverse genetic and environmental factors, leading to mitochondrial damage and subsequent promotion of Parkinson's disease (PD), is attributed to oxidative stress resulting from DA production, despite the relatively moderate energy consumption by the SN. While the authors have conducted a commendable literature study, specific points merit consideration before advancing to the next stage. The following are points and suggestions for your consideration:

1.       Provide more detailed insights into the specific mechanisms that render SNc DA neurons more vulnerable than other neuron types. Please elaborate on the molecular and cellular aspects that contribute to their unique susceptibility.

2.       Elaborate on known factors or characteristics in humans that contribute to the exceptional stress on SNc DA neurons compared to other species. Are there genetic or environmental factors unique to humans that play a role?

3.       Discuss whether the presence of neuromelanin (NM) alone is solely responsible for the observed differences in humans or if additional contributing factors should be considered.

4.       Are there specific markers or indicators that could aid in differentiating whether PD starts in the axonal arbor or the cell bodies of SNc DA neurons? Provide insights into potential diagnostic markers.

5.       Is there experimental evidence or studies supporting the idea that local somatodendritic DA release in human SNc is exceptionally high compared to other species? Provide references or additional details to support this claim.

6.       Discuss specific factors or conditions that exacerbate mitochondrial dysfunction and lead to enhanced PD progression. Highlight any experimental evidence supporting these associations.

7.       What potential therapeutic strategies are explored or suggested to reduce intracellular DA accumulation in SNc DA neuron cell bodies? Provide a comprehensive overview of ongoing research in this area.

8.       Discuss potential reasons for the puzzling lack of success with isradipine in PD treatment. Additionally, suggests alternative avenues for targeting cytosolic Ca2+ oscillations and provides insights into potential challenges.

9.       Elaborate on the role of nutrition and other bioenergetic factors in PD progression. Discuss the significance of these factors in the context of PD risk and potential implications for intervention.

1  Are there alternative pathways or genetic factors that might have been overlooked in the current review? Discuss any additional considerations that could enhance the comprehensiveness of the manuscript.

1 Based on the current findings, what specific avenues for future research do you recommend to enhance further our understanding of PD pathology and potential therapeutic interventions? Provide clear and actionable suggestions.

1Are there specific gaps in the literature that you believe need to be addressed to advance the field? Discuss any limitations or areas where further research is necessary.

Considering these points, I recommend a minor revision to address these considerations further and strengthen the manuscript.

Author Response

We thank the Reviewer for the time and effort to carefully check our paper. We have worked very hard indeed to check available literature and highly appreciate the compliment of us having conducted a commendable literature study.

We also appreciate the kindness of the Reviewer to allow us to expand on several topics of our review, while not making harsh demands on it, as the Reviewer seems well aware that the available information does not allow many solid conclusions. We believe that the changes that we made based on the Reviewer’s requests largely improved the article.

Below, we address each of the individual comments by the Reviewer (in Italic font). Changes that we made to the manuscript are highlighted in its “marked” version.

Comments and Suggestions for Authors

The manuscript entitled "Parkinson’s Disease: Cells Succumbing to Lifelong Dopamine related Oxidative Stress and Other Bioenergetic Challenges” postulates that the susceptibility of substantia nigra pars compacta (SNc) dopamine (DA) neurons to diverse genetic and environmental factors, leading to mitochondrial damage and subsequent promotion of Parkinson's disease (PD), is attributed to oxidative stress resulting from DA production, despite the relatively moderate energy consumption by the SN. While the authors have conducted a commendable literature study, specific points merit consideration before advancing to the next stage. The following are points and suggestions for your consideration:

  1. Provide more detailed insights into the specific mechanisms that render SNc DA neurons more vulnerable than other neuron types. Please elaborate on the molecular and cellular aspects that contribute to their unique susceptibility.

Our response:

We point out that a large part of the article and Fig. 1 were already dedicated to this. However, in line with the Reviewer’s request, we have now added a new figure 2 on dopamine synthesis and its chemistry, and now describe on page 13 (line 470):

“Dopamine is inherently unstable and can produce reactive oxygen species (ROS) through auto-oxidation with metal ions such as Fe3+ as catalysts, and generate DA- and DOPA-quinones (Fig. 2) [128,129]. Quinones are highly reactive molecules that increase oxidative stress [Bolton et al. 2000]. Except for by autoxidation, cytoplasmic DA can also be discarded enzymatically, involving monoamine oxidase (and catechol o-methyl trans-ferase), but this process generates reactive H2O2 (Fig. 2). Thus, if synthesized DA is not de-livered into vesicles and secreted, this leads to oxidative stress within the cytoplasm.”

  1. Elaborate on known factors or characteristics in humans that contribute to the exceptional stress on SNc DA neurons compared to other species. Are there genetic or environmental factors unique to humans that play a role?

Our response:

We have now added a new Table 1 (page 7), titled “Speculations as to why only humans have been found susceptible to idiopathic Parkinson’s disease.”

In that table, we also speculate on a human-specific contribution to PD-vulnerability by “The substantial expansion of the telencephalon, especially the neocortex, in humans imposes a considerable burden on subcortical circuits interacting with the telencephalon (Diederich, N.J.; et al. Mov. Disord. 2019, 34, 453-459) [75].”

However, not much is understood about this human specificity, and therefore we have now also mentioned this topic as “Why is PD only prevalent in humans?” in a new Table 6 (on page 20) dedicated to questions related to SNc DA neuron vulnerability in PD that still need addressing.

  1. Discuss whether the presence of neuromelanin (NM) alone is solely responsible for the observed differences in humans or if additional contributing factors should be considered.

Our response:

See our response to Question #2.

  1. Are there specific markers or indicators that could aid in differentiating whether PD starts in the axonal arbor or the cell bodies of SNc DA neurons? Provide insights into potential diagnostic markers.

Our response:

This is a very complicated question, as these are communicating cellular spaces, plus the definition of “start of neurodegeneration” is not straightforward. Although not precisely following the Reviewer’s suggestion, we have now added a new Table 3 (page 11) titled “Where does the degeneration of SNc DA neurons start, in their cell bodies in the SNc or in their axonal arbor in the striatum?,” in which arguments for either compartment are listed.

  1. Is there experimental evidence or studies supporting the idea that local somatodendritic DA release in human SNc is exceptionally high compared to other species? Provide references or additional details to support this claim.

Our response:

Unfortunately, somatodendritic DA release is an understudied field. Therefore, in the new Table 6 (page 20), one of the open questions that should be addressed in future research is “Do in humans the SNc DA neurons somatodendritically release more DA than in other animals?”

So far, to the best of our knowledge, the only supportive evidence for this is the fact that only humans have abundant neuromelanin, but indeed it should be investigated by direct experiments.

  1. Discuss specific factors or conditions that exacerbate mitochondrial dysfunction and lead to enhanced PD progression. Highlight any experimental evidence supporting these associations.

Our response:

We recognize that we could have delved even deeper into this topic or make more detailed descriptions of the experiments done in this field, but, overall, we believe that with Fig. 1 and a special chapter dedicated to mitochondrial dysfunction, the topic is sufficiently discussed.  

  1. What potential therapeutic strategies are explored or suggested to reduce intracellular DA accumulation in SNc DA neuron cell bodies? Provide a comprehensive overview of ongoing research in this area.

Our response:

That was a great suggestion. We have now added (on page 18) a new chapter “Potential therapies and questions that need addressing” with a new Table 5, titled “Potential therapeutic strategies to reduce stress in SNc DA neuron cell bodies” and a new Table 6, titled “Questions related to SNc DA neuron vulnerability in PD that still need addressing.”

  1. Discuss potential reasons for the puzzling lack of success with isradipine in PD treatment. Additionally, suggests alternative avenues for targeting cytosolic Ca2+ oscillations and provides insights into potential challenges.

Our response:

We have now added to the new chapter mentioned in the response to Question #7, on page 18, line 684, “Given the beauty of the model and the support from studies in rodents and epidemiological data, we also think that the attempts to reduce PD vulnerability by calcium channel blockers like isradipine should be continued. We speculate that this type of drug may also have an adverse effect by blocking DA release and thereby increasing DA toxicity, so that it would be good to test such drug together with a therapy aimed at enhancing DA release (Table 6).”

  1. Elaborate on the role of nutrition and other bioenergetic factors in PD progression. Discuss the significance of these factors in the context of PD risk and potential implications for intervention.

Our response:

To the new Table 5 on possible therapies, we have included a section “Assuring a proper energy supply” and to the new Table 6 on research that we believe should be performed a section “Can the energy status of SNc DA neurons be improved/protected?”

10  Are there alternative pathways or genetic factors that might have been overlooked in the current review? Discuss any additional considerations that could enhance the comprehensiveness of the manuscript.

Our response:

The Reviewer is right that we should also explain the shortcomings of this review. Therefore, on page 21, we have now added a new chapter titled “Limitations of this study.”

11 Based on the current findings, what specific avenues for future research do you recommend to enhance further our understanding of PD pathology and potential therapeutic interventions? Provide clear and actionable suggestions.

Our response:

Again, that was a great request. The urgent questions that we are most interested in are now summarized in the new Table 6 (page 20).

12 Are there specific gaps in the literature that you believe need to be addressed to advance the field? Discuss any limitations or areas where further research is necessary.

Our response:

Although regarding the Reviewer’s suggestion we could have probably done it more systematically and comprehensively, these issues are now being addressed in the new chapters “Potential therapies and questions that need addressing” and “Limitations of this study,” and in the new Table 5 and Table 6.

Considering these points, I recommend a minor revision to address these considerations further and strengthen the manuscript.

Again, we thank the reviewer for the kindness and the many insightful comments and suggestions. We believe that they have resulted in genuine improvements of our article. 

Sincerely,

Also on behalf of Professor Nagatsu and Dr. Dijkstra,

 Hirohisa Watanabe

Hirohisa Watanabe, MD, PhD

Professor, Department of Neurology, Fujita Health University, School of Medicine,

1-98, Dengakugakubo, Kutsukake-cho, Toyoake, Aichi 470-1192, Japan

E-mail: hirohisa.watanabe@fujita-hu.ac.jp

Round 2

Reviewer 2 Report

Comments and Suggestions for Authors

the authors have addressed my concerns. nice work.